# DISSECTING ZERO-SHOT VISUAL REASONING CAPABILITIES IN VISION AND LANGUAGE MODELS

## ABSTRACT

Vision-language models (VLMs) have shown impressive zero- and few-shot performance on real-world visual question answering (VQA) benchmarks, alluding to their capabilities as visual reasoning engines. However, the benchmarks being used conflate "pure" visual reasoning with world knowledge, and also have questions that involve a limited number of reasoning steps. Thus, it remains unclear whether a VLM's apparent visual reasoning performance is due to its world knowledge, or due to actual *visual* reasoning capabilities.

Hence, we systematically benchmark and dissect the zero-shot visual reasoning capabilities of VLMs through synthetic datasets that require minimal world knowledge, and allow for analysis over a broad range of reasoning steps. We focus on two novel aspects of zero-shot visual reasoning: i) evaluating the impact of conveying scene information as either visual embeddings or purely textual scene descriptions to the underlying large language model (LLM) of the VLM, and ii) comparing the effectiveness of chain-of-thought prompting to standard prompting for zero-shot visual reasoning.

We find that the underlying LLMs, when provided textual scene descriptions, consistently perform better compared to being provided visual embeddings. In particular, $\sim$18% higher accuracy is achieved on the PTR dataset. We also find that CoT prompting performs marginally better than standard prompting only for the comparatively large GPT-3.5-Turbo (175B) model, and does worse for smaller-scale models. This suggests the emergence of CoT abilities for visual reasoning in LLMs at larger scales even when world knowledge is limited. Overall, we find limitations in the abilities of VLMs and LLMs for more complex visual reasoning, and highlight the important role that LLMs can play in visual reasoning.

## 1 INTRODUCTION

The development of vision-language models or VLMs (Tan & Bansal, 2019; Li et al., 2020; Wang et al., 2022; Alayrac et al., 2022; Li et al., 2023b; Liu et al., 2023) has gained considerable attention in recent years given their application in developing general-purpose multimodal intelligence. Similar to the zero-shot abilities observed in large language models or LLMs (Brown et al., 2020; Chung et al., 2022; Chowdhery et al., 2022; Touvron et al., 2023) for language tasks, VLMs such as Flamingo (Alayrac et al., 2022) and BLIP-2 (Li et al., 2023b) have shown impressive zero- or few-shot reasoning abilities for language-vision tasks. Notably, they have been shown to surpass task-specific state-of-the-art models (Alayrac et al., 2022; Li et al., 2023b) when finetuned on common visual question answering (VQA) benchmarks, including VQAv2 (Goyal et al., 2017) and OK-VQA (Marino et al., 2019). Furthermore, recent works (Lu et al., 2022; Zhang et al., 2023) have also shown how multimodal chain-of-thought (CoT) reasoning, wherein both language and vision modalities are used to elicit multi-step inference, improves the performance of models on multimodal question answering benchmarks such as ScienceQA (Lu et al., 2022). These findings suggest that similar to LLMs, with increases in model size (Wei et al., 2022a) and advanced prompting techniques (Wei et al., 2022b; Kojima et al.; Jin et al., 2022), VLMs can exhibit stronger reasoning capabilities and operate as instruction-prompted zero- or few-shot visual reasoning engines.

However, the current VQA benchmarks (Goyal et al., 2017; Marino et al., 2019; Hudson & Manning, 2019) used to evaluate the visual reasoning abilities of VLMs predominantly contain questions

requiring only a few reasoning steps, and they often conflate visual reasoning with factual or world knowledge. While open-world visual reasoning certainly relies on knowledge of the world, it is important to recognize that *visual* reasoning at its core encompasses a wide range of cognitive processes including scene interpretation, memory manipulation, spatial reasoning or attention, and logical or semantic inference. To illustrate the above points further, consider the example question "Who is wearing glasses?" (given an image of two individuals) in the popularly-used VQAv2 benchmark. A VLM's accurate answer to this question may simply be due to world knowledge about "glasses" and different categories of "persons", and not necessarily due to better *visual* reasoning capabilities. Similarly, the OK-VQA dataset is particularly designed to test how well models can utilize general world knowledge for VQA, and contains questions such as "What phylum does this animal belong to?" (given an animal image). As such, based on the evaluation benchmarks and analysis in existing works, it is uncertain whether a model's apparent visual reasoning performance is due to its knowledge of the world, or its actual *visual* reasoning capabilities.

Thus, in this work, we propose to systematically analyze and benchmark zero-shot visual reasoning capabilities of VLMs through the usage of synthetic datasets. Specifically, we utilize the CLEVR (Johnson et al., 2017) and PTR (Hong et al., 2021) datasets, which contain questions requiring minimal world knowledge, but a broader range of "reasoning steps" and primitive visual reasoning operations. Moreover, these datasets provide detailed meta-information for each (question, image) pair, including a complete symbolic scene description, as well as a step-by-step functional program for the question. Cumulatively, the broader range of complexities and associated meta-information allow us to better quantify and draw conclusions regarding the "pure" visual reasoning capabilities of VLMs. Additionally, they enable us to assess performance across different fundamental visual operations such as counting, attribute or relationship detection and physical or analogical inferences.

## 1.1 SUMMARY OF EXPERIMENTS AND FINDINGS

We focus on investigating two novel aspects of zero-shot visual reasoning in VLMs. Firstly, we compare the performances of VLMs versus LLMs. Specifically, we compare a "traditional VLM" (i.e. an LLM receiving scene information as visual embeddings from a base vision model) against an LLM simply receiving a completely textual representation of the scene. We find that LLMs consistently outperform VLMs that utilize the same base LLMs. Specifically, in the case of the BLIP2-Flan-T5 (Li et al., 2023b) model, using only its base LLM, i.e. Flan-T5 (Chung et al., 2022), without the visual front-end achieves ~18% higher accuracy on the PTR dataset. One key takeaway is that for questions which can be solved in 2 to 5 "reasoning steps", LLMs show performance levels which are significantly above chance, suggesting that LLMs may in fact possess reasonable capabilities as zero-shot visual reasoning engines.

Secondly, we study how CoT prompting compares to standard prompting for zero-shot application of these models in the context of VQA. We find that CoT prompting for visual reasoning in LLMs only obtains better results than standard prompting at large model scales (in our case for the 175B GPT-3 turbo model) and performs worse for smaller models. For LLMs and VLMs, we observe trends of emergence of CoT reasoning in zero shot settings even when the model's knowledge and context about the world is restricted. Furthermore, owing to the use of synthetic datasets to benchmark VLMs which are not explicitly trained on reasoning on synthetically rendered scenes, we also observe than increase model scale shows signs of improving CoT reasoning capabilities. This indicates that model scaling and CoT could potentially be used to extend and improve zero-shot reasoning performance for multimodal models on previously unseen settings.

## 1.2 CONTRIBUTIONS

(1) To our knowledge, we are the first to systematically benchmark zero-shot visual reasoning capabilities of VLMs using synthetic datasets. This is in order to disentangle the impact of world knowledge, so as to assess the "pure" visual reasoning of models.

(2) We compare the zero-shot VQA performance of VLMs against LLMs, and find that LLMs receiving only ground-truth textual scene information consistently perform better than when provided with visual embeddings.

(3) Consistent with previous studies on CoT for language tasks (Wei et al., 2022b), we find CoT for visual reasoning in LLMs also seems to emerge for larger model sizes even when the model's world knowledge is limited.

(4) We analyze the visual reasoning performance of VLMs and LLMs under various factors including the number of "reasoning steps", question types and model scale. Our overall analysis indicates the limitations of VLMs and LLMs for complex visual reasoning and highlights the important role LLMs can play in enhancing visual reasoning capabilities.

## 2    RELATED WORK

**Benchmarking reasoning capabilities of LLMs and VLMs.**  Since the initial demonstration of LLMs as being effective few-shot learners (Brown et al., 2020), multiple works (Brown et al., 2020; Chung et al., 2022; Zhang et al., 2022; Ouyang et al., 2022; Jin et al., 2022; Chowdhery et al., 2022; Touvron et al., 2023) have sought to refine the design and training of LLMs, besides comprehensively benchmarking (Liang et al., 2022; Srivastava et al., 2022; Valmeekam et al., 2022) their reasoning abilities on language-specific tasks.  More recently, the development of VLMs (Tan & Bansal, 2019; Li et al., 2020; Wang et al., 2022; Alayrac et al., 2022; Li et al., 2023b; Liu et al., 2023) has drawn on advancements in both LLMs and vision-foundation models leading to their prompt-based application for vision-language tasks (Alayrac et al., 2022; Li et al., 2023b; Liu et al., 2023; Wu et al., 2023) such as image captioning, text-guided image editing and general VQA. These works have evaluated the performances of VLMs on prominent VQA benchmarks including VQA-v2 (Goyal et al., 2017), OK-VQA (Marino et al., 2019), GQA (Hudson & Manning, 2019) and VizWiz (Gurari et al., 2018) in zero-shot, few-shot and fine-tuned settings. However, as mentioned before, these analyses are not sufficient to conclude the "true" visual reasoning capabilities of VLMs since the datasets typically conflate world knowledge with visual reasoning and require limited number of "reasoning steps". Further, these works have not assessed whether LLMs by themselves when provided textual (symbolic) scene representations can be capable of visual reasoning in comparison to VLMs. Thus, our work aims to more comprehensively evaluate the zero-shot visual reasoning capabilities of VLMs and their underlying LLMs by utilizing synthetic datasets.

**CoT prompting for zero- or few-shot reasoning.** The development of CoT techniques (Wei et al., 2022b; Kojima et al.; Jin et al., 2022; Yao et al., 2023), wherein models are elicited to reason in multiple steps, has been shown to significantly benefit zero- or few-shot performance of LLMs on diverse language and logical reasoning tasks. More recently, CoT techniques have been developed (Lu et al., 2022; Zhang et al., 2023) to incorporate both vision and language modalities in finetuning LLMs for multimodal question-answering benchmarks such as ScienceQA (Lu et al., 2022).  In contrast to these works, we specifically analyze the impact of CoT prompting in the context of zero-shot VQA for both LLMs and VLMs. Further, to better evaluate how CoT prompting compares with standard prompting for different question types, we provide a breakdown of its performance by the question family for both the PTR and CLEVR datasets.

**Synthetic datasets to disentangle reasoning capabilities from world knowledge**. There are several synthetic datasets which can disentangle world knowledge from reasoning in different ways. (Suhr et al., 2017) is a dataset designed for visual reasoning tasks. The images are synthetic and often involve simple shapes and layouts, ensuring the focus is on reasoning rather than world knowledge. (Kuhnle & Copestake, 2017) generates abstract visual scenes and accompanying textual descriptions designed to test various linguistic and visual phenomena. (Zhang et al., 2019) is a synthetic visual reasoning dataset inspired by the structure of Raven's Progressive Matrices, a popular human IQ test. This format ensures that success on the task requires genuine visual reasoning and pattern recognition, rather than relying on learned associations or world knowledge. (Johnson et al., 2017) and (Hong et al., 2021), the datasets used in this study, are uniquely tailored for disentangling world knowledge from visual reasoning. Their machine-generated questions ensure controlled complexity to test visual reasoning abilities without relying on pre-trained visual or linguistic biases. Their rich annotations and scene metadata are ideal for testing reasoning abilities in VLM as well as LLM models about not only the visual and spatial aspects of the scene, but also the potential physical interactions and outcomes in a wide range of scenarios and reasoning types.

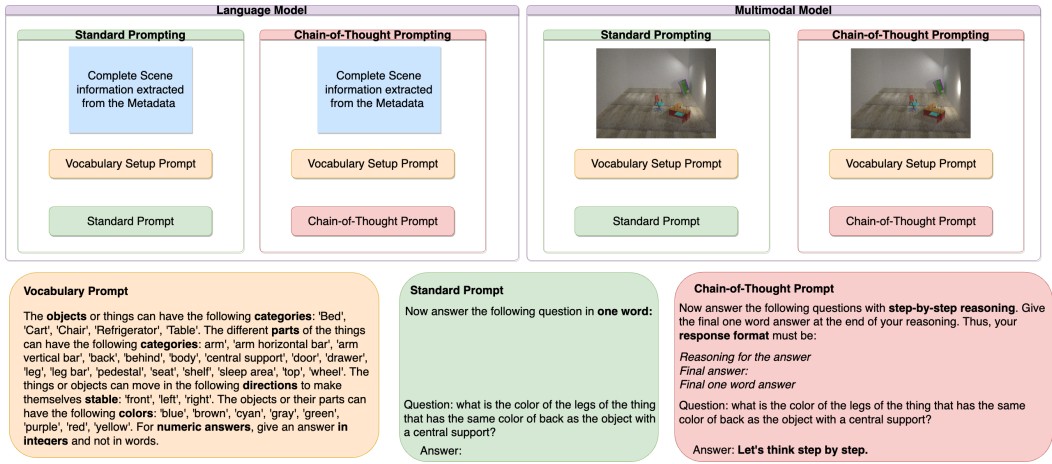

Figure 1: The experimental setup. We perform experiments on pure LLMs as well as their VLM variants with the same set of prompts. In case of LLMs, the image information is provided using the scene metadata used to render the image.

## 3 EXPERIMENTS

### 3.1 EXPERIMENTAL DESIGN

Our experiment design philosophy was primarily guided by the major benchmarks and analysis which we wanted to perform in this study. Our first goal was to analyze the impact of scene information representation in the form of text or images on the model's zero-shot reasoning capabilities. Based on this, we provided the complete scene information in text format to the LLM (the Flan-T5 model family) using the scene metadata, while providing the scene image to the model's VLM counterpart, which was the BLIP-2 Flan-T5 model family (Li et al., 2023b). To gauge the impact of the text-based scene metadata on VLM performance, we also ran a set of experiments providing both the scene metadata and the image to the VLM. Through this setup, we could study areas where the VLM might fall short in terms of information extraction and reasoning, and also identify if there were specific reasoning categories where direct visual representation might be a clear advantage. The second goal was to identify the impact of Chain-of-Thought prompting on the reasoning abilities of LLMs and VLMs as well as its performance trends over scale, when the models world knowledge is limited. To achieve this we designed experiments which could benchmark different scale models of the same LLM and their counterpart VLM families on CoT and Standard Prompts.

### 3.2 EXPERIMENTAL SETUP

Examples of scene metadata and samples of each type of prompt are provided in Appendix A.2.

**Datasets**. We use two datasets: (1) CLEVR (Johnson et al., 2017), a synthetic Visual Question Answering dataset containing images of 3D-rendered objects; each image comes with a number of compositional questions of various types, and (2) PTR (Hong et al., 2021), a dataset for part-based conceptual, relational and physical reasoning. Since the scene metadata was only provided for the images in the train and validating sets (and not the test sets), we use the validation sets of each of these datasets for testing. This allowed us to automatically generate text descriptions of the scenes to compare performance of Visual Language Models (VLMs) with the pure LLMs. There is neither training nor validation per se, since our experiments are in a zero-shot setting.

**Standard prompting**. Our standard prompting procedure included providing the models with the relevant scene information (the image in the case of VLMs, or the scene metadata in the case of pure LLMs), a setup prompt and instructing the model to provide the final answer directly in one word. Since the models were being tested in a purely generative setting, the models would often generate the correct answer, but not use the correct terminology, e.g. calling a cyan object light blue. In order to maintain the generative setting but align the model answers to match the scene terminology, it

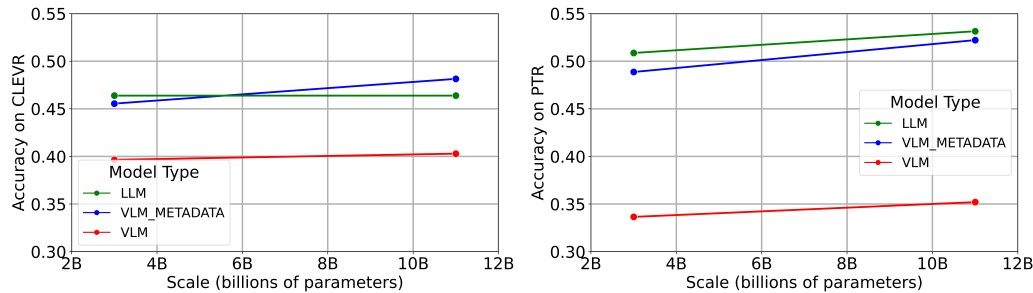

Figure 2: LLM versus VLM+Metadata versus VLM performance on CLEVR and PTR.

was provided with the setup prompt, which gave basic information on the possible attributes, colors, shapes etc which could be present in the scene.

**Chain-of-Thought Prompting**. To elicit CoT reasoning in a zero-shot setting, we follow the prompt template of Kojima et al.. In addition to the same information and setup prompt provided in the standard prompt, we add "Let's think step by step" before each answer. We also developed a format prompt to force the model to give its final one word answer at the end of its reasoning chain.

**Visual Language Models**. We used two VLMs tuned for instructed generation for the experiments. These are BLIP2-Flan-T5-XL (3B) and BLIP2-Flan-T5-XXL (11B). Using the BLIP-2 (Li et al., 2023b) based models allowed us to compare the performance of the VLMs against the pure LLM versions of these models. Pretrained weights from LAVIS (Li et al., 2022) were used.

**Language Models**. We use two LLMs to compare pure language models to VLMs. These are Flan-T5-XL (3B) and Flan-T5-XXL (11B) (Chung et al., 2022). While using the same models at different sizes allowed us to measure the emergent CoT abilities with scale, the true abilities of CoT reasoning have been shown to emerge at a scale of more than 100B. Thus, we also tested our setup on GPT-3.5-Turbo (175B) (Ouyang et al., 2022) and smaller-scale versions of GPT.

## 4 RESULTS AND ANALYSES

### 4.1 COMPARING LLMS WITH SCENE DESCRIPTIONS VERSUS VLMS

**LLMs with scene descriptions outperform VLMs**: Figure 2 shows the impact of visual grounding using BLIP-2 on the reasoning effectiveness of the models. Pure LLMs generally outperform or have similar performance to their counterpart VLM models across both scales and datasets. A t-test was performed to test if the pure LLMs performed better than VLMs. A p-value of 0.0088 indicates that the difference is statistically significant. This might seem counter-intuitive, as one might expect the VLM to be able to effectively utilize the "visual frontend" provided by the image encoder used in the BLIP-2 setup for querying the relevant aspects of the image. There are 2 possible explanations: 1) There are underlying issues in the VLM architecture which prevent the visual front-end from providing relevant information to the model. 2) The complexity of the tasks is not enough that a visual front-end which queries only the relevant information from the scene can be better than providing the complete, unfiltered information to the reasoning engine: which in this case is the LLM. To guard against data contamination (i.e. LLMs trained on CLEVR or PTR), we ran image-free baselines (Appendix A.8), which performed at chance, indicating no contamination.

**LLM advantage for CLEVR versus PTR**: The difference in performance between the LLM and the VLM is more pronounced in PTR than CLEVR. For CLEVR, the LLM outperforms the VLM by roughly 6-7%, while for PTR the gap is roughly 17-18%. One possible explanation is that the objects in PTR are more complex, with multiple parts, hence the task for the VLM's visual frontend is more challenging, and more errors and uncertainty are introduced. Providing the ground-truth scene description to the LLM eliminates this challenging visual frontend task. Conversely, the objects in CLEVR are simple geometric objects, hence access to the ground-truth scene description provides less of an advantage to the LLM.

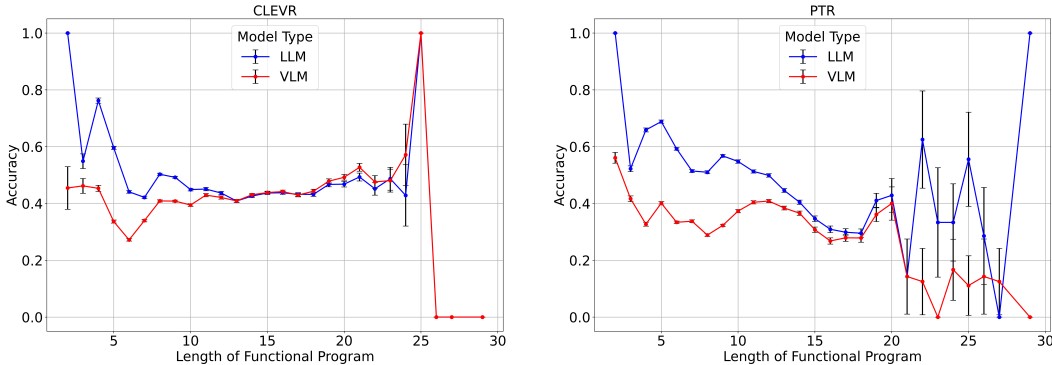

Figure 3: LLM versus VLM performance of Flan-T5-XXL on CLEVR and PTR, analyzed by length of functional programa (a proxy for number of reasoning steps). Error bars represent standard error; large error bars for functional programs longer than 18 are due to the small number of questions.

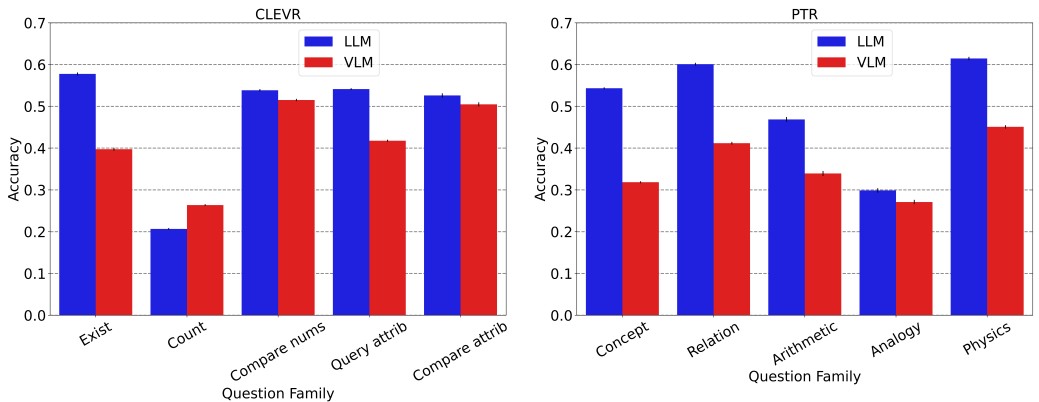

Figure 4: LLM versus VLM model performance of Flan-T5-XXL on CLEVR and PTR using standard prompting, organized by question family.

**Analysis by number of "reasoning steps"**: Both CLEVR and PTR provide functional programs which programmatically describe the solution for the reasoning tasks. We used the length of these functional programs **as a proxy** for the number of "reasoning steps" needed. We analyzed the results by number of "reasoning steps" (Fig. 3). For questions requiring relatively fewer "reasoning steps" (up to around 12-17), LLMs generally outperform VLMs. As seen in Fig. 3 (right), for PTR, both LLMs and VLMs generally show declining performance as the number of "reasoning steps" increases, unsurprisingly. However, when it comes to CLEVR (Fig. 3, left), the performance of VLMs seems to be somewhat independent of the number of "reasoning steps". This could be due to the nature of the CLEVR dataset. CLEVR questions are usually abstract and require deep reasoning, regardless of the number of steps. As such, even tasks with fewer steps might be inherently complex in nature, demanding similar levels of abstraction and reasoning as tasks with more steps.

Moreover, because CLEVR consists of geometric shapes rather than recognizable object parts, the VLMs may not gain as much valuable information from the visual encoder for each additional reasoning step. It is important to note that while the program length provides a heuristic for reasoning complexity, it might not always perfectly capture the cognitive complexity for humans. However, it is still worthwhile to study the impact of length of functional programs on performance.

**Analysis by question family (CLEVR)**: The LLM performs better than the VLM in most categories (Fig. 4, left). The "exist" and "query attribute" categories show the most significant difference in performance, with the LLM noticeably better. Interestingly, the multimodal model performs better in the "count" category. The observed results could potentially be explained by a few factors. For the LLMs, the "exist" and "query attribute" questions are the most straightforward tasks since this

information requires a direct lookup from the scene metadata which already contains this information. The VLMs, on the other hand, require identification of the correct object(s) and their attributes even for "exist" and "query attribute" questions. For "counting" questions, on the other hand, it's possible that VLMs, with their ability to process visual data, are more efficient in tasks like counting where visual cues can be valuable.

**Analysis by question family (PTR)**: The LLM outperforms the VLM across all question families on PTR (Fig. 4, right). The largest performance gap is observed in the "concept" and "relation" categories. "Concept" questions in PTR evaluate a model's capability to understand and reason about basic part-whole relations. Similar to the findings in CLEVR, the question families which require simple "lookups" from the metadata for the LLM have the largest gap in performance. Interestingly, the performance of LLMs on "arithmetic" questions is better than VLMs for this dataset (unlike the "count" questions in CLEVR). This can be attributed to the fact that the level of reasoning required for arithmetic questions is much higher. While such questions in CLEVR were limited to counting objects or comparing numbers, PTR questions require making complex selections of object parts based before performing arithmetic operations.

Visual analogy questions in the PTR dataset require complex reasoning that pose significant challenges for both LLMs and VLMs. This is evident from both the models having their worst performance on the "analogy" question family. This process involves multiple stages of reasoning, including identifying the relevant relationship, applying it to a new context, and generating or selecting the correct answer. The models must not only identify the relationship between A and B, but also accurately project it onto C and D. This complexity could make these tasks particularly challenging for both types of models. Additionally, the geometric and spatial properties involved in analogical reasoning may be difficult for both models.

This question family can also provide insights into the abilities of LLMs to make visual representations of textual descriptions. When provided such a text description of a scene, most humans will try to create a visualization to easily identify the parts or objects which are relevant to the problem at hand. This ability to generate abstract representations from descriptions, or use visual inputs to perform complex projections and analogies still seems to be lacking in existing systems.

**Drawbacks of current VLM Architecture**: VLMs, even those leveraging LLMs, have inherent architectural bottlenecks that may hinder their performance. During inference, they function in two separate phases: 1) visual information querying, where the model's visual frontend extracts scene details based on an initial text query, and 2) text generation, where the LLM uses this extracted information for reasoning and response. This process lacks a feedback loop, preventing the LLM from requesting additional visual information if needed during the generation phase. In contrast, when LLMs receive full scene descriptions in text form, they can access the entire description while generating responses, thereby better retrieving relevant information to answer the question. These drawbacks of VLM architecture are further evidenced by the fact that even when given access to scene metadata, VLMs consistently perform similar to LLMs. This indicates that they are unable to take significant advantage of the additional visual information.

**VLM performance on synthetic vs real images.** One concern of using VLMs on synthetic datasets is that the vision models are not trained on synthetic data, which could lead to lower performance compared to LLMs. We conducted experiments on the GQA (Hudson & Manning, 2019) dataset using a similar LLM vs VLM comparison, and confirmed that the LLMs also performed better than VLMs on natural images. Full analysis and results are in Appendix A.7.

## 4.2 CHAIN-OF-THOUGHT REASONING

**Overall results**: Figure 5 presents a concise summary of the main outcomes of Chain-of-Thought reasoning on the two datasets. Interestingly, the open source Flan-T5-XXL (11B) model with standard prompting achieves the best performance, outperforming even GPT-3.5-Turbo (175B), which is over 15x larger. This is true for both datasets, and regardless of CoT or standard prompting for GPT-3.5-Turbo. Flan-T5-XL (3B) only performed marginally worse than its larger 11B cousin.

**Analysis by number of "reasoning steps"**: As expected, performance generally drops with more "reasoning steps" (Fig. 6). For CLEVR, CoT prompting produced a small but consistent perfor-

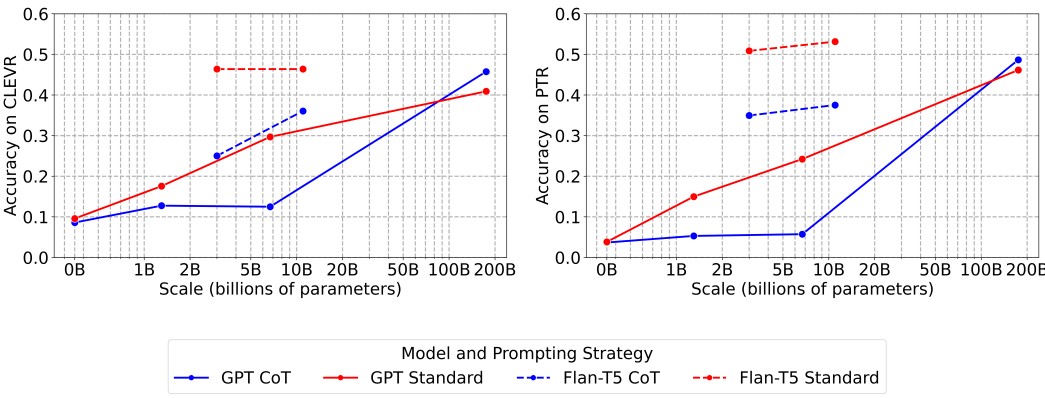

Figure 5: LLM performance on CLEVR and PTR datasets using standard and CoT prompting over scale. The top row represents the GPT models, while the bottom row represents the Flan-T5 models. The x-axis scale is logarithmic for better clarity.

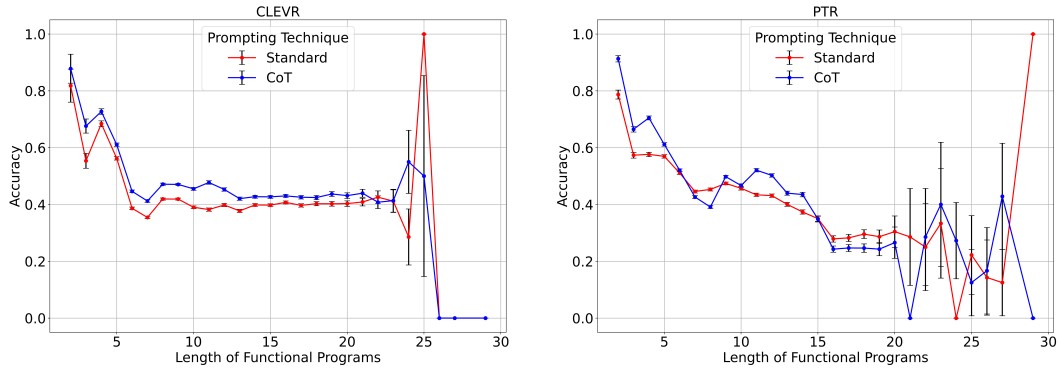

Figure 6: Standard versus CoT prompting performance of GPT-3.5-Turbo on CLEVR and PTR, analyzed by length of functional programs. The vertical black bars indicate standard error bars.

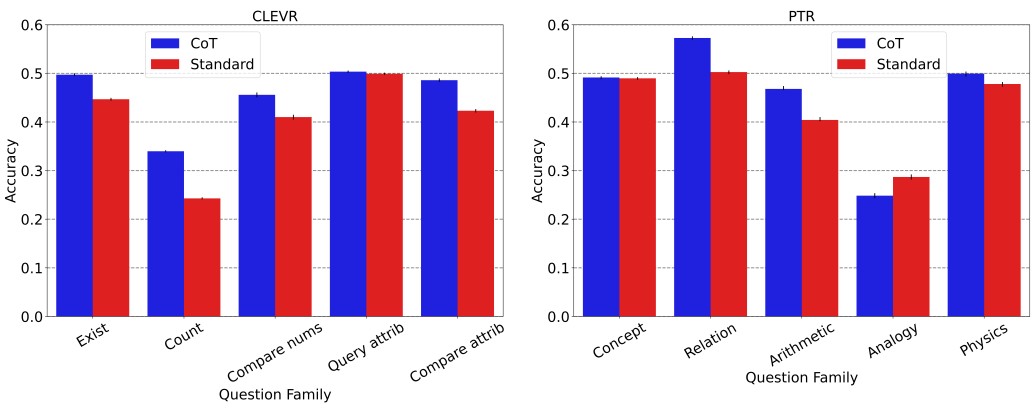

Figure 7: Standard versus CoT prompting performance of GPT-3.5-Turbo on CLEVR and PTR.

mance gain over standard prompting. For PTR, the CoT advantage is less consistent, with standard prompting sometimes performing better.

**Analysis by question family (CLEVR)**: From Fig. 7 (left), CoT prompting shows a noticeable improvement in the "count" question family, with some improvement in "compare attribute", "compare numbers" and "exist" categories. "Query attribute" questions in CLEVR typically involve direct queries about object properties, often solvable in a single step – consistent with the fact that overall accuracy is highest for this question family. This could explain why CoT does not provide a significant advantage in this simple, often one-step question family.

**Analysis by question family (PTR)**: From Fig. 7 (right), CoT prompting leads to improvements for "relation" and "arithmetic" questions. For "analogy" questions, CoT prompting seems to lower performance. CoT prompting assists in "relation" and "arithmetic" questions by breaking down the task into simpler steps, aiding in the understanding of relationships and sequential arithmetic operations. On the other hand, for "analogy" questions, CoT prompting might hinder performance by overly decomposing the problem, possibly losing sight of the overarching relationship.

**Impact of Chain-of-Thought performance across datasets**: CoT prompting resulted in significant improvements in the "count" category in CLEVR and "arithmetic" in PTR, both involving numerical understanding. A possible explanation could be that these tasks are similar to text-based reasoning or step-by-step reasoning examples that the LLMs may have encountered during training. However, the same degree of improvement was not observed in categories such as "analogy" and "query attribute", which are unique to visual reasoning tasks and have no text-based equivalents. The absence of significant improvement in visual reasoning tasks might be due to the fact that base LLMs are not exposed to step-by-step visual reasoning samples or data during training. Consequently, CoT prompting might not be effective for such tasks. This observation could also imply that the generalizability of CoT prompting may be limited. Its effectiveness seems to be largely constrained to tasks that are similar to those the model has previously encountered during training.

**Chain-of-Thought Reasoning over scale:** As seen in Fig. 5, CoT prompting performs better than standard prompting only for a comparatively large GPT-3.5-Turbo (175B) model and does worse for smaller scale models, suggesting the emergence of CoT reasoning at larger scales for visual reasoning tasks, similar to prior observations for other reasoning categories (Wei et al., 2022b).

## 5   LIMITATIONS AND FUTURE WORK

**More varied tasks.** We used datasets for physical reasoning, due to the availability of comprehensive scene metadata and minimal dependency on world knowledge. Future work can extend to a broader range of visual reasoning tasks, such as abstract data interpretation (Kafle et al., 2018), image-based statement classification (Suhr et al., 2017), etc.

**Future work.** We plan to extend our study by benchmarking some of the latest instructed-generation capable VLMs such as Otter (Li et al., 2023a), MultiModal-GPT (Gong et al., 2023) and Instruct-BLIP (Dai et al., 2023) besides recent LLMs such as Chat-GLM (Du et al., 2022), Vicuna (Chiang et al., 2023), OPT (Zhang et al., 2022) and Bloom (Scao et al., 2023) in order to capture trends, bottlenecks and emergent properties for visual reasoning. Additionally, we will benchmark the models on other datasets comprising functional programs such as GQA (Hudson & Manning, 2019) as well as other CoT prompting techniques such as the recent "Tree of Thoughts" (Yao et al., 2023) method.

## 6   CONCLUSION

In this work, we systematically analyzed and benchmarked the zero-shot visual reasoning capabilities of VLMs and LLMs. We specifically utilized synthetic VQA datasets to mitigate the impact of a model's world knowledge on its visual reasoning performance and to also evaluate reasoning over a broader range of "reasoning steps" and primitive visual operations. We studied two novel aspects of zero-shot visual reasoning: i) evaluating how a VLM's base LLM performs when only provided ground-truth textual scene description in comparison to when it is provided with a visual embedding, and ii) comparing the effectiveness of CoT prompting to standard prompting in the context of zero-shot VQA. Further, we extensively analyzed the visual reasoning performance of VLMs and LLMs under various factors, such as number of "reasoning steps", question types and model scale.

## 7 REPRODUCIBILITY STATEMENT

To ensure reproducibility, we have provided a detailed description of the experiment design, as well as experimental setup in section 3. Section A.1 in the appendix contains links to download the relevant datasets, and details on the code submissions as well as general technical documentation to run the experiments.

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

## A APPENDIX

### A.1 EXPERIMENT CODE AND REPRODUCIBILITY

All the relevant code and scripts to process the dataset, run all experiments and evaluate the results is available with the supplemental submission . The code uses 2 major libraries for the experiments:

1. The huggingface transformers library for LLM experiments.
2. The Salesforce-LAVIS library for VLM experiments.

Setup instructions have been included in markdown where required.

The 2 major datasets used (CLEVR, PTR and GQA), can be downloaded from these links:

1. CLEVR

2. PTR

3. GQA

The experiment code can be found in the *code* folder provided along with the supplemental submission. The folder structure is provided in the *README.md* in the root folder and separate files are provided to process the dataset as well as run each experiment for the different model families on different datasets.

## A.2 FULL PROMPT EXAMPLES

## A.3 CLEVR PROMPT EXAMPLE

### A.3.1 IMAGE

The example image used to demonstrate the prompting is provided in figure 8.

### A.3.2 STANDARD PROMPT

Given the following scene:

Scene 0:

Objects: 5 Object: Color: brown Size: large Rotation: 178.92387258999463 Shape: cylinder Material: rubber 3D Coords: [-1.4937210083007812, -1.9936031103134155, 0.699999980079071] Pixel Coords: [119, 131, 10.801968574523926]

Object: Color: gray Size: large Rotation: 243.405459279722 Shape: cube Material: rubber 3D Coords: [1.555708646774292, -2.104736566543579, 0.699999980079071] Pixel Coords: [198, 190, 8.60103988647461]

Object: Color: green Size: small Rotation: 230.45235024165092 Shape: cylinder Material: rubber 3D Coords: [-2.342184543609619, -0.5205014944076538, 0.3499999940395355] Pixel Coords: [161, 118, 12.372727394104004]

Object: Color: purple Size: large Rotation: 31.654351858799153 Shape: sphere Material: metal 3D Coords: [-0.8073106408119202, 1.914123773574829, 0.699999980079071] Pixel Coords: [282, 100, 12.495001792907715]

Object: Color: gray Size: small Rotation: 42.183287560575 Shape: cube Material: metal 3D Coords: [2.6763813495635986, 0.03453871235251427, 0.3499999940395355] Pixel Coords: [337, 195, 9.161211967468262]

Relationships: 'right': [[1, 2, 3, 4], [3, 4], [1, 3, 4], [4], []], 'behind': [[2, 3], [0, 2, 3, 4], [3], [], [0, 2, 3]], 'front': [[1, 4], [], [0, 1, 4], [0, 1, 2, 4], [1]], 'left': [[], [0, 2], [0], [0, 1, 2], [0, 1, 2, 3]]

Directions: 'right': [0.6563112735748291, 0.7544902563095093, -0.0], 'behind': [-0.754490315914154, 0.6563112735748291, 0.0], 'above': [0.0, 0.0, 1.0], 'below': [-0.0, -0.0, -1.0], 'left': [-0.6563112735748291, -0.7544902563095093, 0.0], 'front': [0.754490315914154, -0.6563112735748291, -0.0]

Image Filename: CLEVR_val_000000.png

You may assume that any metal object is shiny, and any rubber object is not shiny ("matte"). All objects are either "metal" or "rubber", and in 2 sizes: "large" or "small". All objects are one of the following colours: "blue", "brown", "cyan", "gray", "green", "purple", "red", "yellow". All objects are one of the following shapes: "cube", "cylinder", "sphere". For numeric answers, give an integer and not in words.

Now answer the following question in one word.

Question: Are there any other things that are the same shape as the big metallic object? Answer:

### A.3.3 CHAIN-OF-THOUGHT PROMPT

Given the following scene:

Scene 0:

Objects: 5 Object: Color: brown Size: large Rotation: 178.92387258999463 Shape: cylinder Material: rubber 3D Coords: [-1.4937210083007812, -1.9936031103134155, 0.699999988079071] Pixel Coords: [119, 131, 10.801968574523926]

Object: Color: gray Size: large Rotation: 243.405459279722 Shape: cube Material: rubber 3D Coords: [1.555708646774292, -2.104736566543579, 0.699999988079071] Pixel Coords: [198, 190, 8.60103988647461]

Object: Color: green Size: small Rotation: 230.45235024165092 Shape: cylinder Material: rubber 3D Coords: [-2.342184543609619, -0.5205014944076538, 0.3499999940395355] Pixel Coords: [161, 118, 12.372727394104004]

Object: Color: purple Size: large Rotation: 31.654351858799153 Shape: sphere Material: metal 3D Coords: [-0.8073106408119202, 1.914123773574829, 0.699999988079071] Pixel Coords: [282, 100, 12.495001792907715]

Object: Color: gray Size: small Rotation: 42.183287560575 Shape: cube Material: metal 3D Coords: [2.6763813495635986, 0.03453871235251427, 0.3499999940395355] Pixel Coords: [337, 195, 9.161211967468262]

Relationships: 'right': [[1, 2, 3, 4], [3, 4], [1, 3, 4], [4], []], 'behind': [[2, 3], [0, 2, 3, 4], [3], [], [0, 2, 3]], 'front': [[1, 4], [], [0, 1, 4], [0, 1, 2, 4], [1]], 'left': [[], [0, 2], [0], [0, 1, 2], [0, 1, 2, 3]]

Directions: 'right': [0.6563112735748291, 0.7544902563095093, -0.0], 'behind': [-0.754490315914154, 0.6563112735748291, 0.0], 'above': [0.0, 0.0, 1.0], 'below': [-0.0, -0.0, -1.0], 'left': [-0.6563112735748291, -0.7544902563095093, 0.0], 'front': [0.754490315914154, -0.6563112735748291, -0.0]

Image Filename: CLEVR_val_000000.png

You may assume that any metal object is shiny, and any rubber object is not shiny ("matte"). All objects are either "metal" or "rubber", and in 2 sizes: "large" or "small". All objects are one of the following colours: "blue", "brown", "cyan", "gray", "green", "purple", "red", "yellow". All objects are one of the following shapes: "cube", "cylinder", "sphere". For numeric answers, give an integer and not in words.

Now answer the following questions with step-by-step reasoning. Give the final one word answer at the end of your reasoning. Thus, your response format should be:

Reasoning for the answer
Final answer:
Final one word answer

Question: Are there any other things that are the same shape as the big metallic object? Answer: Let's think step by step.

## A.4 PTR PROMPT EXAMPLE

### A.4.1 IMAGE

The example image used to demonstrate the prompting is provided in figure 9.

### A.4.2 STANDARD PROMPT

Given the following scene:

Scene PTR_val_007239:

Objects: 4 Object: Category: Chair Rotation: [3.1370301246643066, 0.17649400234222412, 3.115612745285034] Scale: 1.0257009267807007 Stability: no 3D Coords: [4.433284759521484, -6.149937629699707, 0.6772643327713013] Support: [645, 369, 14.225968360900879] Part Colors: 'arm': ['green', [29, 105, 20]], 'back': ['red', [173, 35, 35]], 'central support': ['cyan', [41,

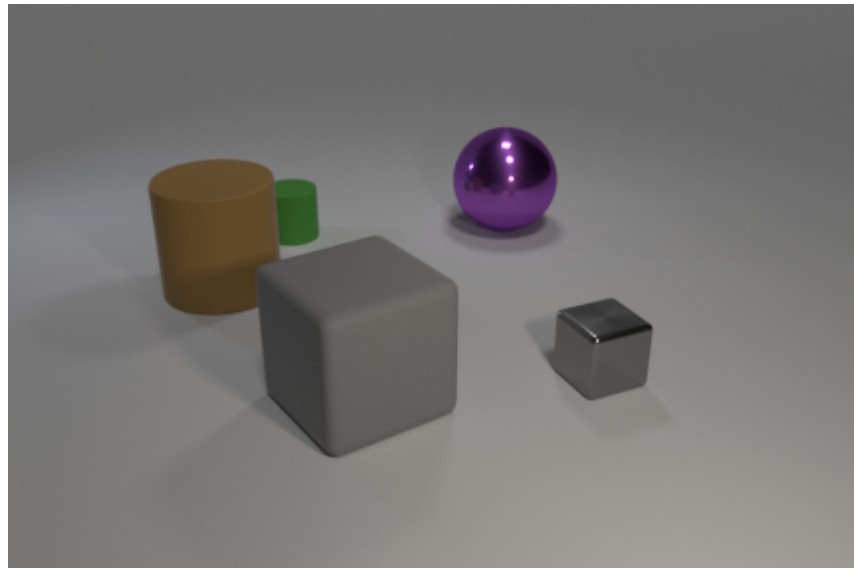

Figure 8: Example CLEVR Image used provided to the Multimodal Models.

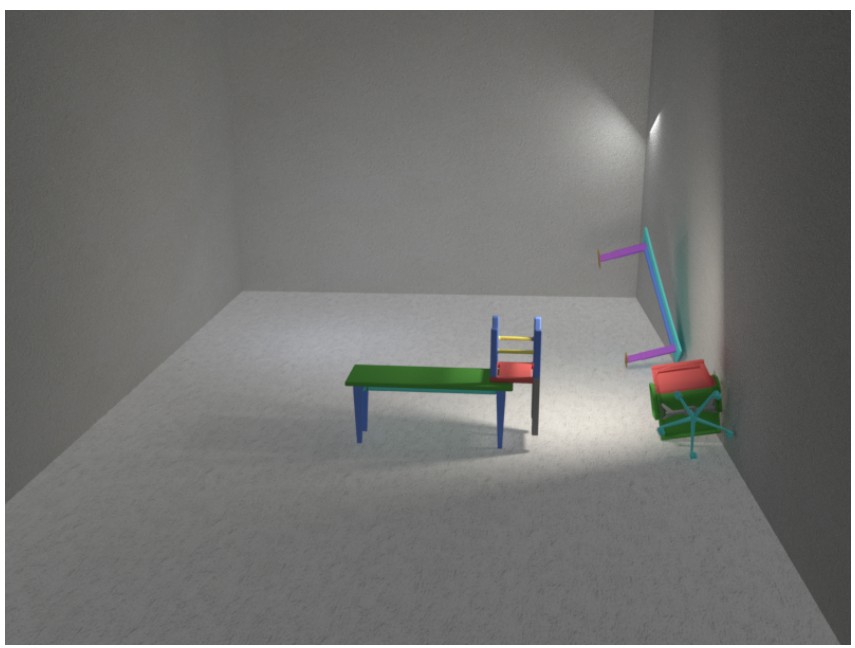

Figure 9: Example PTR Image used provided to the Multimodal Models.

208, 208]], 'leg': ['cyan', [41, 208, 208]], 'seat': ['green', [29, 105, 20]], 'wheel': ['cyan', [41, 208, 208]] Part Count: 'leg': 5, 'wheel': 5

Object: Category: Table Rotation: [1.5707963705062866, -0.0, 3.115298271179199] Scale: 0.7057468096415201 Stability: yes 3D Coords: [0.4326867163181305, -6.359785556793213, 0.8900529742240906] Support: [398, 355, 14.242145538330078] Part Colors: 'leg': ['blue', [42, 75, 215]], 'top': ['green', [29, 105, 20]] Part Count: 'leg': 3

Object: Category: Chair Rotation: [-1.5707963705062866, -0.0, 1.54450201988822021] Scale: 1.0358330011367798 Stability: no 3D Coords: [1.8846343755722046, -6.353758335113525, 1.1621549129486084] Support: [489, 341, 14.07287311553955] Part Colors: 'back': ['gray', [87, 87, 87]], 'leg': ['blue', [42, 75, 215]], 'leg bar': ['yellow', [255, 238, 51]], 'seat': ['red', [173, 35, 35]] Part Count: 'leg': 4, 'leg bar': 2

Object: Category: Table Rotation: [2.6077208518981934, -1.005200743675232, 2.0337021350860596] Scale: 0.6892068386077881 Stability: no 3D Coords: [4.483290195465088, -2.520329236984253, 1.323003888130188] Support: [614, 275, 17.525205612182617] Part Colors: 'leg': ['purple', [129, 38, 192]], 'leg bar': ['brown', [129, 74, 25]], 'top': ['cyan', [41, 208, 208]] Part Count: 'leg': 2, 'leg bar': 2

Relationships: 'above': [[], [2], [], []], 'behind': [[3], [3], [3], []], 'below': [[], [], [1], []], 'front': [[], [], [], [0, 1, 2]], 'left': [[1, 2], [], [], [0, 1, 2]], 'right': [[3], [0, 3], [0, 3], []]

Directions: 'above': [0.0, 0.0, 1.0], 'behind': [-0.05208918824791908, 0.9986424446105957, 0.0], 'below': [-0.0, -0.0, -1.0], 'front': [0.05208918824791908, -0.9986424446105957, -0.0], 'left': [-0.9986424446105957, -0.05208919197320938, 0.0], 'right': [0.9986424446105957, 0.05208919197320938, -0.0]

Image Filename: PTR_val_007239.png

Physics: True

Cam location: [1.4220809936523438, -19.768001556396484, 5.674197196960449]

Cam Rotation: [0.7979968190193176, 0.6015914678573608, 0.02161088027060032, 0.028666317462921143] The objects or things can have the following categories: 'Bed', 'Cart', 'Chair', 'Refrigerator', 'Table'. The different parts of the things can have the following categories: arm', 'arm horizontal bar', 'arm vertical bar', 'back', 'behind', 'body', 'central support', 'door', 'drawer', 'leg', 'leg bar', 'pedestal', 'seat', 'shelf', 'sleep area', 'top', 'wheel'. The things or objects can move in the following directions to make themselves stable: 'front', 'left', 'right'. The objects or their parts can have the following colors: 'blue', 'brown', 'cyan', 'gray', 'green', 'purple', 'red', 'yellow'. For numeric answers, give an answer in integers and not in words.

Now answer the following question in one word.

Question: how many objects are stable? Answer:

### A.4.3 Chain-of-Thought Prompt

Given the following scene:

Scene PTR_val_007239:

Objects: 4 Object: Category: Chair Rotation: [3.1370301246643066, 0.17649400234222412, 3.115612745285034] Scale: 1.0257009267807007 Stability: no 3D Coords: [4.433284759521484, -6.149937629699707, 0.6772643327713013] Support: [645, 369, 14.225968360900879] Part Colors: 'arm': ['green', [29, 105, 20]], 'back': ['red', [173, 35, 35]], 'central support': ['cyan', [41, 208, 208]], 'leg': ['cyan', [41, 208, 208]], 'seat': ['green', [29, 105, 20]], 'wheel': ['cyan', [41, 208, 208]] Part Count: 'leg': 5, 'wheel': 5

Object: Category: Table Rotation: [1.5707963705062866, -0.0, 3.115298271179199] Scale: 0.7057468096415201 Stability: yes 3D Coords: [0.4326867163181305, -6.359785556793213, 0.8900529742240906] Support: [398, 355, 14.242145538330078] Part Colors: 'leg': ['blue', [42, 75, 215]], 'top': ['green', [29, 105, 20]] Part Count: 'leg': 3

Object: Category: Chair Rotation: [-1.5707963705062866, -0.0, 1.5445020198822021] Scale: 1.0358330011367798 Stability: no 3D Coords: [1.8846343755722046, -6.353758335113525, 1.1621549129486084] Support: [489, 341, 14.07287311553955] Part Colors: 'back': ['gray', [87, 87, 87]], 'leg': ['blue', [42, 75, 215]], 'leg bar': ['yellow', [255, 238, 51]], 'seat': ['red', [173, 35, 35]] Part Count: 'leg': 4, 'leg bar': 2

Object: Category: Table Rotation: [2.6077208518981934, -1.005200743675232, 2.0337021350860596] Scale: 0.6892068386077881 Stability: no 3D Coords: [4.483290195465088, -2.520329236984253, 1.323003888130188] Support: [614, 275, 17.525205612182617] Part Colors: 'leg': ['purple', [129, 38, 192]], 'leg bar': ['brown', [129, 74, 25]], 'top': ['cyan', [41, 208, 208]] Part Count: 'leg': 2, 'leg bar': 2

Relationships: 'above': [[], [2], [], []], 'behind': [[3], [3], [3], []], 'below': [[], [], [1], []], 'front': [[], [], [], [0, 1, 2]], 'left': [[1, 2], [], [], [0, 1, 2]], 'right': [[3], [0, 3], [0, 3], []]

Directions: 'above': [0.0, 0.0, 1.0], 'behind': [-0.05208918824791908, 0.9986424446105957, 0.0], 'below': [-0.0, -0.0, -1.0], 'front': [0.05208918824791908, -0.9986424446105957, -0.0], 'left': [-0.9986424446105957, -0.05208919197320938, 0.0], 'right': [0.9986424446105957, 0.05208919197320938, -0.0]

Image Filename: PTR_val_007239.png

Physics: True

Cam location: [1.4220809936523438, -19.768001556396484, 5.674197196960449]

Cam Rotation: [0.7979968190193176, 0.6015914678573608, 0.02161088027060032, 0.028666317462921143] The objects or things can have the following categories: 'Bed', 'Cart', 'Chair', 'Refrigerator', 'Table'. The different parts of the things can have the following categories: arm', 'arm horizontal bar', 'arm vertical bar', 'back', 'behind', 'body', 'central support', 'door', 'drawer', 'leg', 'leg bar', 'pedestal', 'seat', 'shelf', 'sleep area', 'top', 'wheel'. The things or objects can move in the following directions to make themselves stable: 'front', 'left', 'right'. The objects or their parts can have the following colors: 'blue', 'brown', 'cyan', 'gray', 'green', 'purple', 'red', 'yellow'. For numeric answers, give an answer in integers and not in words.

Now answer the following questions with step-by-step reasoning.Give the final one word answer at the end of your reasoning. Thus, your response format should be:

Reasoning for the answer
Final answer:
Final one word answer

Question: how many objects are stable? Answer: Let's think step by step.

## A.5 Prompt For VLM with full scene metadata

In the case of VLMs with full scene metadata, we use the same exact scene description as provided for Standard prompts, with the image input used the same as in VLM experiments.

## A.6 Full Experimental Results

The results for all experiments performed are given in the Table 1

### A.6.1 VLM CoT Performance Discussion

The decision to omit the VLM CoT results from the main paper was mainly due to space constraints and our initial assessment that these results might not offer as much insight as the other findings. We aimed to streamline the main content of the paper for clarity and conciseness.

Here are the results and discussion for the VLM CoT Prompting performance:

**Observations**

For both the CLEVR and PTR datasets, the accuracy of BLIP-2 Flan-T5 XXL models is generally higher than the BLIP-2 Flan-T5 XL models, regardless of the prompting technique. Across both

Table 1: Experiment Results

| Model | Scale (Billions of parameters) | Dataset | Type | Prompting Technique | Accuracy |
|---|---|---|---|---|---|
| FLAN T5 | 3.00 | CLEVR | LLM | CoT | 0.250172 |
| BLIP-2 FLAN T5 | 3.00 | CLEVR | Multimodal | CoT | 0.138455 |
| FLAN T5 | 3.00 | CLEVR | LLM | Standard | 0.463932 |
| BLIP-2 FLAN T5 | 3.00 | CLEVR | Multimodal | Standard | 0.396497 |
| BLIP-2 FLAN T5 | 3.00 | CLEVR | Multimodal + Full Scene Metadata | Standard | 0.455474 |
| FLAN T5 | 11.00 | CLEVR | LLM | CoT | 0.360632 |
| BLIP-2 FLAN T5 | 11.00 | CLEVR | Multimodal | CoT | 0.282964 |
| FLAN T5 | 11.00 | CLEVR | LLM | Standard | 0.463932 |
| BLIP-2 FLAN T5 | 11.00 | CLEVR | Multimodal | Standard | 0.402938 |
| BLIP-2 FLAN T5 | 11.00 | CLEVR | Multimodal + Full Scene Metadata | Standard | 0.481456 |
| GPT | 0.35 | CLEVR | LLM | CoT | 0.085992 |
| GPT | 0.35 | CLEVR | LLM | Standard | 0.095729 |
| GPT | 1.30 | CLEVR | LLM | CoT | 0.127561 |
| GPT | 1.30 | CLEVR | LLM | Standard | 0.175713 |
| GPT | 6.70 | CLEVR | LLM | CoT | 0.124974 |
| GPT | 6.70 | CLEVR | LLM | Standard | 0.296915 |
| GPT | 175.00 | CLEVR | LLM | CoT | 0.457613 |
| GPT | 175.00 | CLEVR | LLM | Standard | 0.409037 |
| FLAN T5 | 3.00 | PTR | LLM | CoT | 0.349455 |
| BLIP-2 FLAN T5 | 3.00 | PTR | Multimodal | CoT | 0.071239 |
| FLAN T5 | 3.00 | PTR | LLM | Standard | 0.508657 |
| BLIP-2 FLAN T5 | 3.00 | PTR | Multimodal | Standard | 0.336524 |
| BLIP-2 FLAN T5 | 3.00 | PTR | Multimodal + Full Scene Metadata | Standard | 0.488672 |
| FLAN T5 | 11.00 | PTR | LLM | CoT | 0.375339 |
| BLIP-2 FLAN T5 | 11.00 | PTR | Multimodal | CoT | 0.228783 |
| FLAN T5 | 11.00 | PTR | LLM | Standard | 0.531447 |
| BLIP-2 FLAN T5 | 11.00 | PTR | Multimodal | Standard | 0.352028 |
| BLIP-2 FLAN T5 | 11.00 | PTR | Multimodal + Full Scene Metadata | Standard | 0.522143 |
| GPT | 0.35 | PTR | LLM | CoT | 0.036834 |
| GPT | 0.35 | PTR | LLM | Standard | 0.038419 |
| GPT | 1.30 | PTR | LLM | CoT | 0.053044 |
| GPT | 1.30 | PTR | LLM | Standard | 0.149849 |
| GPT | 6.70 | PTR | LLM | CoT | 0.057316 |
| GPT | 6.70 | PTR | LLM | Standard | 0.242263 |
| GPT | 175.00 | PTR | LLM | CoT | 0.486693 |
| GPT | 175.00 | PTR | LLM | Standard | 0.461586 |

datasets and both model sizes, the Standard Prompting technique consistently outperforms the CoT (Chain of Thought) prompting technique. The performance drop due to CoT is more pronounced in the smaller BLIP-2 Flan-T5 XL models compared to the larger XXL variants. For example, in the CLEVR dataset, the XL model shows a drop of 25.8% (from 39.65% to 13.85%) when using CoT compared to Standard Prompting, while the XXL variant shows a smaller drop of 12% (from 40.29% to 28.29%). A similar trend is observed in the PTR dataset where the performance drop in the XL model is 26.53% (from 33.65% to 7.12%) compared to a drop of 12.32% (from 35.20% to 22.88%) in the XXL model when switching from Standard Prompting to CoT.

**Analysis**

**Prompting Technique Influence.** While Standard Prompting seems to be the more effective method across both datasets, the Chain of Thought (CoT) reasoning does show potential, and trends of emergence over scale. This is especially important considering that the VLMs are not explicitly trained on synthetic images, suggesting that CoT emergence in VLMs is not limited to the tasks or image categories for which they were trained. Additionally, it provides further evidence for the observation that CoT does seem to emerge even in the absence of world knowledge.

**Implications for Future Research.** While scale evidently improves performance, there's a need to further investigate the interaction between prompting techniques and model scale. The larger drop in performance in smaller models when using CoT indicates that certain reasoning capabilities emerge more robustly at higher scales. Future research could delve deeper into optimizing prompting techniques specifically for smaller models or further enhancing the performance of larger models.

A.7  GQA EXPERIMENTS

The GQA dataset was used to test the experimental setup on a dataset which uses natural images instead of synthetically generated images. This was done in order to check the fairness of the VLM vs LLM comparision on the original datasets. The rationale behind this was that the Visual encoders in the VLMs were not trained on synthetic images, which affect the performance on the datasets selected in the original paper. The GQA dataset was as it provided access to comprehensive scene metadata as well as functional programs to arrive at the answer, similar to the (Johnson et al., 2017) and (Hong et al., 2021) datasets used in the main experiments. To facilitate our analysis, we used annotated scene graphs as a proxy for perfect scene information. Further, since GQA contains 1469 labels which can hinder a model's effectiveness (when providing the label vocabulary through prompting), we drew a subset of questions for the top-25 labels and used a maximum prompt length of 20,000 tokens. This resulted in 78536 questions out of the original 132062 questions in the official validation set. The code to sample the dataset and run the experiments is provided along with the supplemental submission A.1. we observe that the LLM (Flan-T5-XXL) has accuracy of 78.72% while the VLM (BLIP2-Flan-T5-XXL) with the same base language model has accuracy of 56.81%. The overall model performance is provided in Table 2, performance over length of functional programs is provided in Table 3 and the performance over the question families is provided in Table 4.

**Analysis of the results.** We can see that the LLM outperforms the VLM on the dataset, as well as over the length of functional programs and question families. This result is consistent with the findings of the main paper. It is important to note that there are not many questions with a large length of the functional programs in the dataset, the scene metadata covers all the important relationships and informations in a more verbose manner and the answers seem to be generally simpler to answer than the synthetic datasets, which could explain a relatively larger gap in the LLM vs VLM performance.

Table 2: Experiment Results on Sampled GQA Dataset

| Model | Dataset | Accuracy |
|---|---|---|
| Flan-T5 XXL | Sampled GQA Dataset | 78.72 |
| Blip-2 Flan-T5 XXL | Sampled GQA Dataset | 56.81 |

Table 3: Performance Over Length of Functional Programs on GQA

| Length | Model | Correct | Total | Accuracy (%) |
|---|---|---|---|---|
| 2 | Flan-T5 XXL | 20404 | 26977 | 75.63 |
| 2 | Blip-2 Flan-T5 XXL | 14517 | 26977 | 53.81 |
| 3 | Flan-T5 XXL | 23921 | 29574 | 80.88 |
| 3 | Blip-2 Flan-T5 XXL | 16914 | 29574 | 57.19 |
| 4 | Flan-T5 XXL | 6685 | 8100 | 82.53 |
| 4 | Blip-2 Flan-T5 XXL | 4920 | 8100 | 60.74 |
| 5 | Flan-T5 XXL | 7819 | 10369 | 75.40 |
| 5 | Blip-2 Flan-T5 XXL | 6538 | 10369 | 63.05 |
| 6 | Flan-T5 XXL | 77 | 83 | 92.77 |
| 6 | Blip-2 Flan-T5 XXL | 56 | 83 | 67.47 |
| 7 | Flan-T5 XXL | 2908 | 3426 | 84.88 |
| 7 | Blip-2 Flan-T5 XXL | 1664 | 3426 | 48.57 |
| 8 | Flan-T5 XXL | 6 | 6 | 100.0 |
| 8 | Blip-2 Flan-T5 XXL | 5 | 6 | 83.33 |
| 9 | Flan-T5 XXL | 1 | 1 | 100.0 |
| 9 | Blip-2 Flan-T5 XXL | 1 | 1 | 100.0 |

A.7.1  GQA EXAMPLE PROMPT

**Image** The example image used to demonstrate the prompting for GQA is provided in 10

**Standard Prompt**

Table 4: Performance Over Question Families on GQA

| Question Family | Model | Correct | Total | Accuracy (%) |
|---|---|---|---|---|
| Logical | Flan T5 XXL | 12503 | 15590 | 80.19 |
| Logical | Blip-2 Flan T5 XXL | 9829 | 15590 | 59.58 |
| Verify | Flan T5 XXL | 21145 | 26355 | 80.23 |
| Verify | Blip-2 Flan T5 XXL | 15967 | 26355 | 60.58 |
| Query | Flan T5 XXL | 18027 | 21910 | 82.27 |
| Query | Blip-2 Flan T5 XXL | 12669 | 21910 | 57.82 |
| Choose | Flan T5 XXL | 7585 | 11374 | 66.68 |
| Choose | Blip-2 Flan T5 XXL | 4925 | 11374 | 45.30 |
| Compare | Flan T5 XXL | 2561 | 3307 | 77.44 |
| Compare | Blip-2 Flan T5 XXL | 1765 | 3307 | 53.27 |

context: Given the following scene: Image Dimensions: 500x347 Objects: 34 Object ID 1231798: Name: face Coordinates: x=442, y=55 Dimensions: w=16, h=25 Attributes: Relation: Name: of Object: 1231760 Relation: Name: to the right of Object: 1231777 Relation: Name: to the right of Object: 1231778

Object ID 1231799: Name: face Coordinates: x=68, y=20 Dimensions: w=25, h=36 Attributes: Relation: Name: to the right of Object: 1231788 Relation: Name: of Object: 1231770 Relation: Name: to the left of Object: 1231781 Relation: Name: to the left of Object: 1231773

Object ID 1231790: Name: cart Coordinates: x=168, y=141 Dimensions: w=168, h=192 Attributes: Relation: Name: to the right of Object: 1231781 Relation: Name: to the right of Object: 1231780 Relation: Name: to the right of Object: 1231770 Relation: Name: to the right of Object: 1231779 Relation: Name: to the right of Object: 1231797 Relation: Name: to the right of Object: 1231796 Relation: Name: to the left of Object: 1231760 Relation: Name: to the right of Object: 1231768 Relation: Name: to the right of Object: 1231767 Relation: Name: to the left of Object: 1231763

Object ID 1231792: Name: outlet Coordinates: x=199, y=338 Dimensions: w=13, h=7 Attributes: Relation: Name: on Object: 1231793

Object ID 1231793: Name: floor Coordinates: x=0, y=167 Dimensions: w=498, h=180 Attributes:

Object ID 1231794: Name: arm Coordinates: x=43, y=76 Dimensions: w=70, h=48 Attributes: Relation: Name: to the left of Object: 1231781 Relation: Name: to the right of Object: 1231786 Relation: Name: to the left of Object: 1231802 Relation: Name: to the left of Object: 1231783

Object ID 1231796: Name: leg Coordinates: x=96, y=184 Dimensions: w=31, h=104 Attributes: Relation: Name: of Object: 1231770 Relation: Name: to the left of Object: 1231781 Relation: Name: to the right of Object: 1231797 Relation: Name: to the left of Object: 1231790 Relation: Name: to the left of Object: 1231767 Relation: Name: to the left of Object: 1231766

Object ID 1231797: Name: leg Coordinates: x=13, y=187 Dimensions: w=56, h=123 Attributes: Relation: Name: to the left of Object: 1231796 Relation: Name: to the left of Object: 1231768 Relation: Name: of Object: 1231770 Relation: Name: to the left of Object: 1231767 Relation: Name: to the left of Object: 1231766 Relation: Name: to the left of Object: 1231781 Relation: Name: to the left of Object: 1231790

Object ID 1231800: Name: shirt Coordinates: x=20, y=56 Dimensions: w=95, h=136 Attributes: Relation: Name: to the left of Object: 1231802 Relation: Name: to the left of Object: 1231781 Relation: Name: to the left of Object: 1231764 Relation: Name: to the left of Object: 1231779 Relation: Name: to the right of Object: 1231786

Object ID 1231802: Name: shirt Coordinates: x=113, y=57 Dimensions: w=60, h=93 Attributes: Relation: Name: to the left of Object: 1231783 Relation: Name: to the left of Object: 1231764 Relation: Name: to the right of Object: 1231800 Relation: Name: to the right of Object: 1231794 Relation: Name: to the right of Object: 1231786 Relation: Name: to the left of Object: 1231782 Relation: Name: to the left of Object: 1231777 Relation: Name: to the left of Object: 1231778

Object ID 1231769: Name: sandal Coordinates: x=2, y=304 Dimensions: w=31, h=38 Attributes: brown

Object ID 1231768: Name: sandal Coordinates: x=96, y=286 Dimensions: w=51, h=26 Attributes: brown Relation: Name: to the left of Object: 1231790 Relation: Name: to the right of Object: 1231797

Object ID 1231761: Name: arm Coordinates: x=430, y=85 Dimensions: w=40, h=64 Attributes: Relation: Name: to the right of Object: 1231762 Relation: Name: to the right of Object: 1231782 Relation: Name: to the right of Object: 1231777

Object ID 1231760: Name: girl Coordinates: x=370, y=48 Dimensions: w=107, h=224 Attributes: Relation: Name: to the right of Object: 1231777 Relation: Name: carrying Object: 1231762 Relation: Name: to the right of Object: 1231790 Relation: Name: to the right of Object: 1231764 Relation: Name: to the right of Object: 1231783 Relation: Name: to the right of Object: 1231782 Relation: Name: with Object: 1231762 Relation: Name: to the right of Object: 1231778

Object ID 1231763: Name: sneakers Coordinates: x=427, y=251 Dimensions: w=27, h=20 Attributes: white Relation: Name: to the right of Object: 1231790

Object ID 1231762: Name: purse Coordinates: x=392, y=114 Dimensions: w=52, h=55 Attributes: Relation: Name: to the right of Object: 1231777 Relation: Name: to the right of Object: 1231764 Relation: Name: to the right of Object: 1231782 Relation: Name: to the left of Object: 1231761

Object ID 1231764: Name: projector Coordinates: x=196, y=126 Dimensions: w=74, h=30 Attributes: Relation: Name: to the right of Object: 1231800 Relation: Name: to the right of Object: 1231802 Relation: Name: to the left of Object: 1231760 Relation: Name: to the left of Object: 1231762 Relation: Name: to the left of Object: 1231782 Relation: Name: to the right of Object: 1231781 Relation: Name: to the left of Object: 1231777 Relation: Name: to the right of Object: 1231770 Relation: Name: to the right of Object: 1231779

Object ID 1231767: Name: sandal Coordinates: x=134, y=246 Dimensions: w=31, h=19 Attributes: brown Relation: Name: to the left of Object: 1231790 Relation: Name: to the right of Object: 1231770 Relation: Name: to the right of Object: 1231797 Relation: Name: to the right of Object: 1231796

Object ID 1231766: Name: sandal Coordinates: x=164, y=237 Dimensions: w=39, h=12 Attributes: brown Relation: Name: to the right of Object: 1231780 Relation: Name: to the right of Object: 1231797 Relation: Name: to the right of Object: 1231796 Relation: Name: to the right of Object: 1231770

Object ID 1231788: Name: books Coordinates: x=41, y=21 Dimensions: w=9, h=19 Attributes: Relation: Name: to the left of Object: 1231799 Relation: Name: to the left of Object: 1231771 Relation: Name: to the left of Object: 1231773 Relation: Name: to the left of Object: 1231781

Object ID 1231783: Name: wall Coordinates: x=223, y=0 Dimensions: w=72, h=120 Attributes: brick Relation: Name: to the right of Object: 1231781 Relation: Name: to the left of Object: 1231760 Relation: Name: to the left of Object: 1231777 Relation: Name: to the right of Object: 1231787 Relation: Name: to the right of Object: 1231802 Relation: Name: to the right of Object: 1231794 Relation: Name: to the right of Object: 1231770 Relation: Name: to the left of Object: 1231778 Relation: Name: to the right of Object: 1231773

Object ID 1231782: Name: game Coordinates: x=268, y=135 Dimensions: w=40, h=18 Attributes: Relation: Name: to the right of Object: 1231770 Relation: Name: to the right of Object: 1231779 Relation: Name: to the left of Object: 1231762 Relation: Name: to the right of Object: 1231764 Relation: Name: to the right of Object: 1231781 Relation: Name: to the right of Object: 1231802 Relation: Name: to the left of Object: 1231761 Relation: Name: to the left of Object: 1231760

Object ID 1231781: Name: people Coordinates: x=112, y=24 Dimensions: w=88, h=240 Attributes: Relation: Name: to the right of Object: 1231788 Relation: Name: to the left of Object: 1231783 Relation: Name: to the left of Object: 1231764 Relation: Name: to the right of Object: 1231787 Relation: Name: to the right of Object: 1231786 Relation: Name: to the left of Object: 1231782 Relation: Name: to the right of Object: 1231780 Relation: Name: to the left of Object: 1231777 Relation: Name: in front of Object: 1231786 Relation: Name: wearing Object: 1231779 Relation:

Name: to the right of Object: 1231774 Relation: Name: to the left of Object: 1231778 Relation: Name: to the right of Object: 1231770 Relation: Name: playing Object: 1231782 Relation: Name: to the right of Object: 1231794 Relation: Name: to the left of Object: 1231790 Relation: Name: to the right of Object: 1231799 Relation: Name: to the right of Object: 1231800 Relation: Name: to the right of Object: 1231797 Relation: Name: to the right of Object: 1231796 Relation: Name: wearing Object: 1231780 Relation: Name: wearing Object: 1231802

Object ID 1231780: Name: shorts Coordinates: x=22, y=175 Dimensions: w=106, h=74 Attributes: Relation: Name: to the left of Object: 1231779 Relation: Name: to the left of Object: 1231766 Relation: Name: to the left of Object: 1231790 Relation: Name: to the left of Object: 1231781

Object ID 1231787: Name: shelf Coordinates: x=16, y=7 Dimensions: w=97, h=39 Attributes: brown Relation: Name: to the left of Object: 1231781 Relation: Name: to the left of Object: 1231783 Relation: Name: by Object: 1231786 Relation: Name: to the left of Object: 1231773

Object ID 1231786: Name: window Coordinates: x=1, y=76 Dimensions: w=20, h=58 Attributes: Relation: Name: to the left of Object: 1231781 Relation: Name: behind Object: 1231781 Relation: Name: to the left of Object: 1231794 Relation: Name: to the left of Object: 1231802 Relation: Name: to the left of Object: 1231800

Object ID 1231776: Name: hair Coordinates: x=432, y=49 Dimensions: w=22, h=17 Attributes: Relation: Name: to the right of Object: 1231778 Relation: Name: to the right of Object: 1231777

Object ID 1231777: Name: girl Coordinates: x=274, y=42 Dimensions: w=60, h=194 Attributes: Relation: Name: to the right of Object: 1231802 Relation: Name: to the right of Object: 1231764 Relation: Name: to the left of Object: 1231762 Relation: Name: to the left of Object: 1231760 Relation: Name: to the right of Object: 1231783 Relation: Name: to the right of Object: 1231781 Relation: Name: to the left of Object: 1231776 Relation: Name: to the right of Object: 1231779 Relation: Name: to the left of Object: 1231798 Relation: Name: to the left of Object: 1231761

Object ID 1231774: Name: beard Coordinates: x=73, y=49 Dimensions: w=17, h=12 Attributes: Relation: Name: to the left of Object: 1231781 Relation: Name: to the left of Object: 1231773

Object ID 1231773: Name: hair Coordinates: x=124, y=25 Dimensions: w=30, h=35 Attributes: Relation: Name: to the right of Object: 1231774 Relation: Name: to the right of Object: 1231799 Relation: Name: to the left of Object: 1231783 Relation: Name: to the right of Object: 1231787 Relation: Name: to the right of Object: 1231788

Object ID 1231770: Name: man Coordinates: x=3, y=7 Dimensions: w=145, h=333 Attributes: Relation: Name: to the left of Object: 1231779 Relation: Name: to the left of Object: 1231764 Relation: Name: to the left of Object: 1231790 Relation: Name: wearing Object: 1231762 Relation: Name: to the left of Object: 1231766 Relation: Name: to the left of Object: 1231767 Relation: Name: wearing Object: 1231800 Relation: Name: wearing Object: 1231780 Relation: Name: to the left of Object: 1231781 Relation: Name: to the left of Object: 1231782 Relation: Name: to the left of Object: 1231783 Relation: Name: with Object: 1231774

Object ID 1231771: Name: hair Coordinates: x=48, y=7 Dimensions: w=48, h=33 Attributes: Relation: Name: to the right of Object: 1231788

Object ID 1231778: Name: hair Coordinates: x=286, y=42 Dimensions: w=38, h=43 Attributes: Relation: Name: to the right of Object: 1231802 Relation: Name: to the left of Object: 1231760 Relation: Name: to the right of Object: 1231783 Relation: Name: to the right of Object: 1231781 Relation: Name: to the left of Object: 1231798 Relation: Name: to the left of Object: 1231776

Object ID 1231779: Name: shorts Coordinates: x=119, y=133 Dimensions: w=59, h=83 Attributes: Relation: Name: to the left of Object: 1231790 Relation: Name: to the right of Object: 1231770 Relation: Name: to the right of Object: 1231800 Relation: Name: to the left of Object: 1231764 Relation: Name: to the right of Object: 1231780 Relation: Name: to the left of Object: 1231782 Relation: Name: to the left of Object: 1231777

The possible answers could be: yes, no, left, right, man, white, black, bottom, woman, blue, chair, top, brown, table, boy, gray, bed, green, girl, red, cat, dog, car, bus, horse.Now answer the following question in one word. Question: What color is the helmet in the middle of the image?

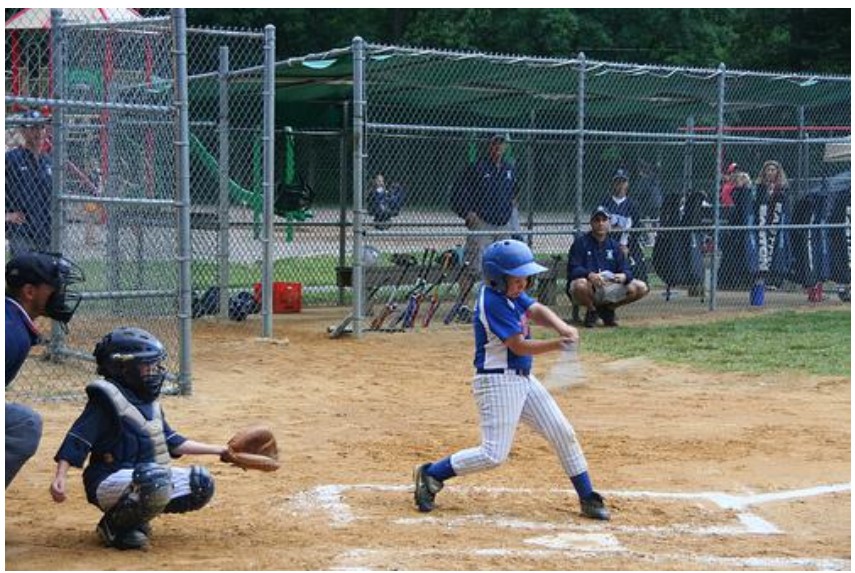

Figure 10: Example GQA Image used provided to the VLM Models.

### A.8 IMAGE-FREE BASELINE AND RANDOM CHANCE

We conducted image-free experiments for GPT 3.5 models on both CLEVR and PTR to establish a baseline to which the model performance could be compared. This meant providing the **Dataset name, dataset split (val) and image name,** followed by **answer vocabulary hint** to the model, similar to the original prompt, and then asking the question, **without providing any scene metadata**. Since the GPT 3.5 model was trained on the open internet, there is a chance that it could have seen some of the dataset during its training process. Establishing an image-free baseline enabled us to gauge whether the model had prior information about the questions and the scenes.

For Flan-T5, the datasets on which the model was trained have been disclosed and do not contain CLEVR or PTR. More details about Flan-T5 training and fine-tuning is available in Appendix F of the main paper.

The code required to run the experiments have been provided with the supplemental submission under the "image_free" folder

**Image-free baseline prompt – CLEVR**

Answer the following question from the val split of the CLEVR Dataset for image CLEVR_val_000000.png You may assume that any metal object is shiny, and any rubber object is not shiny ("matte"). All objects are either "metal" or "rubber", and in 2 sizes: "large" or "small". All objects are one of the following colours: "blue", "brown", "cyan", "gray", "green", "purple", "red", "yellow". All objects are one of the following shapes: "cube", "cylinder", "sphere". For numeric answers, give an integer and not in words. Always answer the following question in a single word from the options provided above. Your response should only be a single word. Question: Is there a big brown object of the same shape as the green thing?

Answer:

**Image-free baseline prompt – PTR**

Answer the following question from the val split of the PTR Dataset for image PTR_val_007239.png The objects or things can have the following categories: 'Bed', 'Cart', 'Chair', 'Refrigerator', 'Table'. The different parts of the things can have the following categories: arm', 'arm horizontal bar', 'arm vertical bar', 'back', 'behind', 'body', 'central support', 'door', 'drawer', 'leg', 'leg bar', 'pedestal', 'seat', 'shelf', 'sleep area', 'top', 'wheel'. The things or objects can move in the following directions to make themselves stable: 'front', 'left', 'right'. The objects or their parts can have the following colors: 'blue', 'brown', 'cyan', 'gray', 'green', 'purple', 'red', 'yellow'. For numeric

answers, give an answer in integers and not in words. Always answer the following question in a single word from the options provided above. Your response should be just a single word. Question : how many objects are stable?

Answer:

**Image-free baseline – Results**

The models response for most such questions would be that "**The question cannot be answered without more information**". Thus, we forced a valid response from the model by asking it to always give a one word answer from the answer vocabulary provided.

*CLEVR Image-free Baseline*

The Image free baseline performance of GPT 3.5 on CLEVR was **36.85%**. Table 5 calculates the random chance of getting a question from CLEVR right. We see that the image free baseline results indicate that the model performance in the absence of scene metadata is basically random chance.

Table 5: Random Chance and Total Questions for CLEVR

| Category | Random Chance (%) | Total Questions |
|---|---|---|
| exist | 50.00 | 20196 |
| colors | 12.50 | 13404 |
| material | 50.00 | 30545 |
| compare attribute | 50.00 | 35422 |
| shape | 33.33 | 13544 |
| size | 50.00 | 10094 |
| count | 10.00 | 13273 |
| compare numbers | 50.00 | 13513 |
| Overall random chance | | 36.86 |

*PTR Image-free Baseline*

The Image free baseline performance of GPT 3.5 on CLEVR was **10.16%**. Table 6 calculates the random chance of getting a question from CLEVR right. Again, we see that the image free baseline results indicate that the model performance in the absence of scene metadata is basically random chance.

Table 6: Random Chance and Total Questions for PTR

| Category | Random Chance (%) | Total Questions |
|---|---|---|
| concept | 2.63 | 38972 |
| relation | 4.35 | 22905 |
| physics | 50.00 | 7413 |
| analogy | 5.26 | 7472 |
| arithmetic | 8.33 | 14958 |
| Overall random chance | | 8.03 |

## A.9 COMPUTE USED

The models were trained using different compute resources depending on the scale. All Multimodal models were run on NVIDIA A100 (40GB) VRAM GPUs. For the FLAN-T5 family, the 11B models used NVIDIA A100 (40GB), while the smaller ones (3B) used NVIDIA A40's. Complete details can be found in 7.

Table 7: Compute Used for Different Models

| Model | Compute Used | Experiment (Both CLEVR and PTR) |
|---|---|---|
| BLIP-2 FLAN-T5 (3B) | NVIDIA A40 | Multimodal CoT |
| BLIP-2 FLAN-T5 (3B) | NVIDIA A40 | Multimodal Standard |
| FLAN-T5 (3B) | NVIDIA A40 | LLM CoT |
| FLAN-T5 (3B) | NVIDIA A40 | LLM Standard |
| BLIP-2 FLAN-T5 (3B) | NVIDIA A40 | Multimodal CoT |
| BLIP-2 FLAN-T5 (3B) | NVIDIA A40 | Multimodal Standard |
| FLAN-T5 (3B) | NVIDIA A40 | LLM CoT |
| FLAN-T5 (3B) | NVIDIA A40 | LLM Standard |
| BLIP-2 FLAN-T5 (11B) | NVIDIA A100 (40GB) | Multimodal CoT |
| BLIP-2 FLAN-T5 (11B) | NVIDIA A100 (40GB) | Multimodal Standard |
| FLAN-T5 (11B) | NVIDIA A100 (40GB) | LLM CoT |
| FLAN-T5 (11B) | NVIDIA A100 (40GB) | LLM Standard |
| BLIP-2 FLAN-T5 (11B) | NVIDIA A100 (40GB) | Multimodal CoT |
| BLIP-2 FLAN-T5 (11B) | NVIDIA A100 (40GB) | Multimodal Standard |
| FLAN-T5 (11B) | NVIDIA A100 (40GB) | LLM CoT |
| FLAN-T5 (11B) | NVIDIA A100 (40GB) | LLM Standard |
| GPT (175B) | OpenAI API | LLM CoT |
| GPT (175B) | OpenAI API | LLM Standard |
| GPT (175B) | OpenAI API | LLM CoT |
| GPT (175B) | OpenAI API | LLM Standard |
| GPT (350M) | OpenAI API | LLM CoT |
| GPT (350M) | OpenAI API | LLM Standard |
| GPT (350M) | OpenAI API | LLM CoT |
| GPT (350M) | OpenAI API | LLM Standard |
| GPT (1.3B) | OpenAI API | LLM CoT |
| GPT (1.3B) | OpenAI API | LLM Standard |
| GPT (1.3B) | OpenAI API | LLM CoT |
| GPT (1.3B) | OpenAI API | LLM Standard |
| GPT (6.7B) | OpenAI API | LLM CoT |
| GPT (6.7B) | OpenAI API | LLM Standard |
| GPT (6.7B) | OpenAI API | LLM CoT |
| GPT (6.7B) | OpenAI API | LLM Standard |

## A.10 EXAMPLES OF LIMITATIONS

### A.10.1 GIBBERISH ANSWERS

We will use the full example for the CLEVR dataset provided in A.3.3 to demonstrate some commonly types of non-sensical answers which the models provide.

**Repeating the Question** The following question was provided to the Flan-T5 3B parameter LLM model with the chain-of-thought prompt provided in section A.3.3 as scene description.

*Question*: There is a small gray block; are there any spheres to the left of it?

*Model Response*: There is a small gray block; are there any spheres to the right of it?

Here the model just repeats the question provided to it.

**Repeating the input prompt** The following question was provided to the Flan-T5 3B parameter LLM model with the chain-of-thought prompt provided in section A.3.3 as scene description.

*Question*: What is the color of the large shiny sphere?

*Model Response*: You may assume that any metal object is shiny, and any rubber object is not shiny ("matte"). All objects are either metallic or rubbery, and in 2 sizes: "large" or "small". All objects are one of the following colours: "blue", "brown", "cyan", "gray", "green", "purple", "red",

"yellow". All objects are one of the following shapes: "cube", "cylinder", "sphere". For numeric answers, give an integer and not in words.

Here the model simply repeats the setup prompt which was used to align its responses with the dataset answer vocabulary.

**Repeating the same logic over and over**

*Question*: Are there any other things that are the same shape as the big metallic object?

*Model Response*: The big metallic object is a cube. The small metallic object is a cube. The purple object is a cube. The red object is a cube. The yellow object is a cube. The blue object is a cube. The brown object is a cube. The green object is a cube. The red object is a cube. The yellow object is a cube. The blue object is a cube. The brown object is a cube. The purple object is a cube. The red object is a cube. The yellow object is a cube. The blue object is a cube. The brown object is a cube. The green object is a cube. The purple object is a cube. The red object is a cube. The yellow object is a cube. The blue object is a cube. The brown object is a cube. The purple object is a cube. The yellow object is a cube. The red object is a cube. The brown object is a cube. The green object is a cube. The yellow object is a cube. The blue object is a cube. The brown object is a cube. The purple object is a cube. The yellow object is a cube. The blue object is a cube. The brown object is a cube. The purple object is a cube. The yellow object is a cube. The blue object is a cube. The brown object is a cube. The purple object is a cube. The yellow object is a cube. The blue object is a cube. The brown object is a cube. The purple object is a cube. The yellow object is a cube. The blue object is a cube. The brown object is a cube. The purple object is a cube. The yellow object is a cube. The blue object is a cube. The brown object is a cube. The purple object is a cube. The yellow object is a cube. The blue object is a cube. The brown object is a cube. The purple object

As seen here, the model gets lost in its own explanations and keeps repeating the same logic over and over.

**Not adhere to the response format** The following question was provided to the Flan-T5 3B parameter LLM model with the chain-of-thought prompt provided in section A.3.3 as scene description.

*Question*: Is there a big brown object of the same shape as the green thing?

*Model Response*: The green object is a cube, and the brown object is a cube too

As seen here, while the model response is correct, it does not follow the response format provided to it.

### A.10.2 PROMPT ENGINEERING

Prompt engineering plays an important role in eliciting the required reasoning behaviour from the instruction based LLMs used in the project. Changes in the wording of the prompt, or emphasis on different aspects of the instructions can have drastic changes in the model response.

**Guidance for Response Structure**. We noticed the most impact in model responses on the answer formatting aspect of the prompt. The larger scale models were significantly better in following a required format for their response, while the smaller models required further emphasis for the same. For example, the smaller model variants of both the GPT as well as Flan-T5 models adhered to the required format much better when the response format was repeated at the end of the prompt as well.

**Impact of prompt format**. The formatting of the input prompt affects the model response as well. The Flan-T5 models are text-to-text models which have been trained on question answering in a specific format. During their training, they were made to elicit CoT reasoning for their responses by providing samples in the following format:

Example of the ideal prompt
<Scene metadata + setup prompt>
*Question*: <Question about the provided scene>
*Answer*: Lets think step-by-step.

From our observations, the Flan-T5 models were able to provide best responses for CoT reasoning when provided the prompt in this exact format. Other formats for the prompts, such as those given below, were not as effective in making the models perform CoT reasoning.

Examples of ineffective prompts

*Example 1*
<Scene metadata + setup prompt>
Answer the following question with step-by-step reasoning:
*Question*: <Question about the provided scene>
*Answer*:

*Example 2*
<Scene metadata + setup prompt>
Answer the following question with step-by-step reasoning:
*Question*: <Question about the provided scene>
Lets think step-by-step.

This indicates that while the models can exhibit general reasoning abilities, their performance is not yet flexible or robust to prompts they might not have observed in their training data before. Their performance for the same task still heavily depends on the format and content of the training data.

**Observations**: From empirical observations, it seems like the models become more robust to prompt format and better at following instructions with scale, since the models of the same type show progressive improvements in both these aspects as scale increases.

