# OpenReview forum: "Dissecting Zero-Shot Visual Reasoning Capabilities in Vision and Language Models"
_ICLR.cc/2024/Conference — ICLR 2024 Conference Withdrawn Submission_

### Official Review · Reviewer_PwJg · 2023-10-21

**Soundness:** 2 fair
**Presentation:** 2 fair
**Contribution:** 2 fair
**Rating:** 3
**Confidence:** 4

**Summary:**

This paper systematically studies and compares the performance of LLMs and VLMs on visual reasoning problems using two synthetic datasets. The main findings are: 1. LLMs provided with ground-truth textual scene information perform better than those provided with visual embeddings. 2. CoT only helps when the scale of the LLMs is large enough. 3. They also analyze how the number of “reasoning steps”, question types, and model scale affect the performance.

**Strengths:**

1. This is the first work to test VLMs and LLMs in visual reasoning tasks. It performs detailed experiments and analysis on both VLMs and LLMs models.
2. The study is meaningful and provides some reference for both research in visual reasoning tasks and LLMs and LVLMs.

**Weaknesses:**

1. The paper only studies one LVLMs (BLIP-2 FlanT5), the conclusions may not generalized to more recent LVLMs such as LLAVA and InstructBLIP.
2. Some results and analysis are not very inspiring: E.g. 1. It is reasonable and not surprising that for synthetic visual reasoning tasks, providing ground truth metadata to LLMs is better than visual embedding processed by the model since ground truth metadata contains all the required information to reason the answer. 2. The observation that larger LLMs perform better using chain-of-thought has been discovered by prior work [1].
3. The conclusion in the paragraph **Drawbacks of current VLM Architecture** is not well supported by the experiments. Instead, they are more like some hypothesis.

[1] Chain-of-Thought Prompting Elicits Reasoning in Large Language Models

**Questions:**

1. When providing explanations for the experiment result, the reviewer think it is better to provide some qualitative examples to support it. E.g. 'One possible explanation is that the objects in PTR are more complex, with multiple parts, hence the task for the VLM’s visual frontend is more challenging, and more errors and uncertainty are introduced.'

---

### Official Review · Reviewer_13QZ · 2023-10-30

**Soundness:** 2 fair
**Presentation:** 2 fair
**Contribution:** 2 fair
**Rating:** 5
**Confidence:** 4

**Summary:**

This article systematically examine and benchmark the zero-shot visual reasoning capabilities of VLMs through synthetic datasets that require minimal world knowledge, and allow for analysis over a broad range of reasoning steps.

**Strengths:**

In my opinion, the most captivating aspect of this article is systematically examine and benchmark the zero-shot visual reasoning capabilities of VLMs through synthetic datasets that require minimal world knowledge. It can effectively separates actual visual reasoning capabilities from vast amounts of world knowledge obtained in VLM pre-training. Furthermore, the experimental strategy design is reasonable.

**Weaknesses:**

While the initial insight is intriguing, as mentioned above, the experimental section falls short in effectively supporting the current viewpoints and uncovering more compelling conclusions. However, in the Chapter 5 LIMITATIONS AND FUTURE WORK, authors demonstrate a clear understanding of their weaknesses and identify the improvements of their research.

**Questions:**

As mentioned above, I suggest that the authors further explore more varied visual reasoning tasks and evaluate the latest extensive VLM models on more datasets unrelated to world knowledge. While the initial insight is valid and the experimental strategy design is soundness, the execution and argumentation are lacking. If the authors address the points raised in the Chapter 5 and conduct a more comprehensive analysis to determine if VLM possesses genuine visual reasoning capabilities, I believe this work will generate significant interest within the VLM community. However, at its current stage, I find the work to be insufficient.

---

### Official Review · Reviewer_An5H · 2023-11-01

**Soundness:** 1 poor
**Presentation:** 2 fair
**Contribution:** 3 good
**Rating:** 3
**Confidence:** 3

**Summary:**

The paper is a straightforward investigation of two things. First, whether VLMs can outperform the corresponding LLMs they bootstrap from. Second, whether chain-of-thought prompting (“let’s think step by step…”) helps LLMs on structured visual understanding tasks (on CLEVR and PTR).

**Strengths:**

**Writing**:
- The writing is easy to follow which is great.
- The related work section is mostly thorough.

**Results of interest**:
1. That VLM performance could be lower than LLM performance on a visual task is somewhat surprising.
2. In the CoT analysis: that Flan-T5 outperforms GPT-3.5 consistently is also surprising.

**Weaknesses:**

**Criticisms/questions**:
1. Are the VLMs and LLMs directly comparable? The VLMs only bootstrap from the pretrained LLMs. But after training, there’s not much reason for them to be comparable.
2. If they are directly comparable, why is VLM performance lower than LLM performance even when the VLMs have the scene metadata? They should be able to zero-out the contribution of the visual modality in each case. That should permit them to perform as well as language-only reasoning.
3. Figure 2: to study scaling laws we need more than 2 datapoints. Given the BLIP family only offers two models, it could be useful to consider other model families. We'll also need error bars.

**Writing**:
- The paper could use some more sign-posting (e.g., there’s nothing to introduce Section 4).
- Some sections are really large (e.g., Section 4.1) and could be structured better.
- Figure captions could be better/more informative (e.g., Figure 5 refers to a “top row” and “bottom row” which are hard to find.)
- With some compression (e.g., figures) the paper could fit 7 pages perhaps.

**Questions:**

Please see weaknesses above, which I've tried to frame as questions.

---

### Official Review · Reviewer_Jf6s · 2023-11-02

**Soundness:** 2 fair
**Presentation:** 2 fair
**Contribution:** 2 fair
**Rating:** 3
**Confidence:** 4

**Summary:**

The paper analyzes Vision-language models (VLMs) and large language models (LLMs) on zero-shot visual reasoning. To minimize the impact of background knowledge on the inference result, the authors chose synthetic datasets including CLEVR and PTR. The main findings are: (1) LLMs are better than VLMs on average; (2) chain-of-thought is helpful only on 175B parameter GPT model.

**Strengths:**

- The choice of using synthetic data to focus on reasoning ability is well-motivated.
- Performance analysis based on reasoning step, question family, model size, sheds some light on the behavior of VLMs and LLMs.
- Performance was measured on GPT and FLAN-T5 across different parameter scale.

**Weaknesses:**

- The paper has some interesting observations and hypothesis. However, majority of them are not backed up by experiments.
For example, for the claim "visual information querying, where the model’s visual frontend extracts scene
details based on an initial text query," more evidence is appreciated. If this is indeed the case, the authors should try modifying initial text query to extract richer visual information. Another ways is leveraging a feedback loop as authors metioned as done in works like:
(1) Haoxuan You et al., "IdealGPT: Iteratively Decomposing Vision and Language Reasoning via Large Language Models" (2) Kaiwen Zhou et al., "ViCor: Bridging Visual Understanding and Commonsense Reasoning with Large Language Models"

Authors also should explore alternative possibilities such as scene information extracted by VLMs being incorrect.
- Lacking insights. The authors classified the problems by # reasoning steps, question family (exist, count, etc). Why and how LLMs fail on these categories? Also, how LLMs compare on these categories compared to more traditional methods such as neuro-symbolic reasoners for CLEVR dataset? Are there any characteristic of visual reasoning problems that poses challenges to VLMs?
- Only one kind of VLM is used. Hence, the conclusion from the current results may not generalize to other VLMs.
- Writing and presentation could have been significantly improved. For example, figures could be condensed into numerical numbers in Tables.

**Questions:**

- Please answer first two points on Weaknesses section.